# Quality of newborn care and associated factors: An analysis of the 2022 Kenya demographic and health survey

John Baptist Asiimwe[1]*, Earnest Amwiine[2], Angella Namulema[3], Quraish Sserwanja[4], Joseph Kawuki[5], Mathius Amperiize[6], Shamim Nabidda[1], Imelda Namatovu[7], Lilian Nuwabaine[1]*

1 Aga Khan University, Uganda Campus, Kampala, Uganda, 2 Faculty of Medicine, Mbarara University of Science and Technology, Mbarara, Uganda, 3 Mbarara Regional Referral Hospital, Mbarara, Uganda, 4 Programs Department, Relief International, Khartoum, Sudan, 5 Department of Family, Population, & Preventive Medicine, Stony Brook University, Stony Brook, New York, United States of America, 6 Infectious Diseases Institute, Kampala, Uganda, 7 Makerere University, Kampala, Uganda

* john.asiimwe@aku.edu (JBA); lilliannuwabaine@gmail.com (LN)

**Data Availability Statement:** Third party data was obtained for this study from The DHS Program (https://dhsprogram.com/). Data may be requested

## Abstract

Kenya one of the African countries has pledged to reduce neonatal death as per the 2030 World Health Organization target. Providing high-quality newborn care is critical in minimizing neonatal mortality. This study aimed to determine the factors that influence the quality of newborn care in Kenya. Secondary data from 11,863 participants of the 2022 Kenya Demographic and Health Survey (KDHS) were analyzed. The participants were chosen using two-stage stratified sampling. The quality of newborn care was operationalized as receiving all components of newborn care after childbirth, as reported by the mother. Using SPSS (version 29), univariate and multivariable logistic regression analyses were used to analyse the data. In this study, 32.7% (95% confidence interval [CI]: 31.0%-34.5%) of the mothers reported that their newborns had received all components of quality neonatal care after childbirth. Mothers who spent an average of one hour accessing the health facilities compared with those who spent less than half an hour were 1.33 (95%CI: 1.01–1.75) times more likely to report that their newborns had received quality newborn care. Mothers who gave birth in a non-government organization health facility were 30.37 (95%CI: 2.69–343.20) times more likely to report that their newborns had received quality newborn care compared with those who delivered from a faith-based organization. On the contrary, in terms of regions, mothers who lived in Nyanza, Eastern, and Rift Valley provinces compared with those who lived in the coastal regions were 0.53 (95%CI: 0.34–0.82), 0.61 (95%CI: 0.39–0.94), and 0.62 (95%CI: 0.41–0.93) times less likely to report that their newborns had received quality newborn care, respectively. Mothers who subscribed to other religions or faith (0.28 (95%CI: 0.10–0.76) compared with those from the Christian faith, were less likely to report that their newborns had received quality newborn care. Finally, mothers who gave birth through cesarean section were 0.44 (95%CI: 0.32–0.61) times less likely to report that their newborns had received quality newborn care than mothers who gave birth through spontaneous vaginal delivery. The study indicates that about a third of the neonates received quality newborn care and that facility-related and parental social factors were

from The DHS Program after creating an account and submitting a concept note. More access information can be found on The DHS Program website (https://dhsprogram.com/data/Access-Instructions.cfm). The data set is openly available upon permission from the MEASURE DHS website (https://www.dhsprogram.com/data/available-datasets.cfm). The authors confirm that interested researchers would be able to access these data in the same manner as the authors. The authors also confirm that they had no special access privileges that others would not have.

**Funding:** The authors received no specific funding for this work.

**Competing interests:** The authors have declared that no competing interests exist.

associated with receiving quality newborn care. Stakeholders need to pay more attention to newborn babies whose mothers come from certain regions of Kenya where the quality of newborn care was found to be low, minority religious faith denominations, and those who delivered by ceasearen section. Stakeholders also should focus on strengthening collaborations with NGO health facilities and achieving universal health coverage to improve the quality of newborn care provided in health facilities.

## Introduction

According to the World Health Organization (WHO), the quality of newborn care is the extent to which newborn health care services meet the desired health outcomes of newborns and must be provided in a safe, timely, and appropriate manner while putting into consideration the needs (or preferences) of newborns and their families [1]. Quality newborn care is critical to assisting babies adapt to the extrauterine environment and can reduce neonatal death by 25% [2]. WHO emphasizes the importance of ensuring quality care for neonates during the postpartum period as recommended in the form of assessing the newborn's temperature, checking the umbilical cord, counseling, and observing the mother during breastfeeding [3]. The other components of quality neonatal care include the management of hypothermia, neonatal sepsis and other medical conditions, birth from a skilled attendant and the health facility, examination of the baby, skin-to-skin care, and immunization [4,5].

Despite the decline in the under-five mortality worldwide, deaths remains concentrated in the very first days of life indicating the need to attend to quality neonatal care demands [6]. The WHO estimated approximately 2.4 million newborn deaths in 2019, which accounted for over 45% of under-five deaths worldwide. Additionally, a persistent slow reduction in newborn mortality was also reported in the East African region [7]. The first 28 days of life present a very crucial period when babies are most vulnerable to health challenges and face a higher risk of death with the global mortality rate standing at 17 deaths per 1,000 live births in 2022 [8]. However, the rate is higher in sub-Saharan countries (Kenya inclusive), which contributes 57% of the worldwide mortality rate, with rates of 21 deaths per 1000 live births in low and middle-income countries [9–11]. All these mortality rates exceed the projected sustainable development goal target of 12 deaths for every 1000 live births by 2030 [12].

The high newborn mortality rate is attributed to poor-quality newborn care [13–15]. Providing quality newborn care has been reported to be low in African countries like Kenya where the neonatal mortality rate is still a healthcare challenge, accounting for 40% of under-five mortality rates [16–19]. In Sub-Saharan Africa, challenges to delivering newborn care are multi-faceted and include poor healthcare infrastructure, social-cultural-environmental challenges, lack of adherence to clinical guidelines, and inadequate political will by governments [20–22]. The WHO guidelines on postnatal care emphasize the provision of quality newborn care as a means of reducing preventable neonatal mortality [23].

Evidence from the literature indicates that the level of education, monthly income, age of the mother, number of antenatal visits, and knowledge of newborn care by the mothers were some of the factors associated with the quality of newborn care [24–26]. Other sociocultural determinants of the quality of newborn care include residency, maternal employment, pregnancy intention, health facility accessibility, occupation, parity, and counseling during the perinatal period [27–30]. Furthermore, healthcare system-related factors affecting the quality of neonatal care include the presence of health workers at work stations, access to evidence-based guidelines, working environment, communication skills, and supervision [31].

To achieve the sustainable development target 3.2 of eliminating unnecessary deaths of newborns and children under the age of five, there is a need to invest in monitoring the quality of newborn care delivered by healthcare providers in countries with limited resources like Kenya where neonatal mortality is still high [23]. Some studies that have been used to draw recommendations in Kenya like prioritizing the need to strengthen existing health facilities and staffing, increasing access to standard inpatient neonatal care, and developing effective referral systems to improve neonatal care [32,33], were not conducted on a nationally representative sample population. Therefore, this study aimed to identify the parental, health facility, obstetric, and child-related factors associated with the quality of newborn care (i.e., receiving all the newborn care services) as reported by the mothers using nationally representative data from Kenya. Identifying these factors is crucial in developing efficient interventions to combat neonatal mortality and morbidity in Kenya and sub-Saharan Africa at large.

## Methods

### Data source, sample design, and collection

This study used data from the 2022 Kenya Demographic and Health Survey (KDHS). The survey used a two-stage stratified sampling approach in which the first step involved using equal probabilities and random selection to choose 1692 clusters from a master sample frame of 129,067 clusters taken from the 2019 Kenya population and housing census. The second stage entailed house listing to create a sampling frame whereby 25 households for each cluster were chosen. If a cluster contained fewer than 25 households, all of them were sampled. In total, 1691 clusters were surveyed. The Inner-City Fund (ICF) aided with the training of data collectors and pre-testing of study materials. Data was collected from February to July 2022.

All women between 15 and 49 years who were either regular members of the selected families or had resided in those homes the night preceding the survey were interviewed about newborn care in English or Swahili. Out of 32,156 women who responded to the questionnaire, 11,863 had recently given birth within the previous five years [11]. The MEASURE DHS website (https://www.dhsprogram.com/data/available-datasets.cfm) gave permission to utilize the 2022 KDHS dataset. Although the dataset comprised several variables, only those that were relevant to our research were included in the analysis.

### Study variables

**Dependent/Outcomes.** The main outcome variable in this study was the mothers' reported quality of newborn care. The quality of newborn care was an aggregate variable estimated from several binary outcomes (yes or no), which included whether the newborn received certain services from medical professionals during or after birth, as reported by their mother. The services included obtaining delivery assistance from a skilled health provider, weighing the newborn, taking the newborn's temperature, placing the newborn on the mother's breast within the first hour after delivery, observing the mother while breastfeeding their newborns, teaching the mother ways to clean the baby's umbilical cord, examining the newborn's umbilical cord, advising the mothers regarding risk indicators of neonatal illness (danger signs), and guiding or counseling the mother on breastfeeding. As reported by the mother, receiving all the newborn care services implied that one had received quality newborn care (yes) or not (no) if one skipped 1 of the 9 services ($\geq 1$, binary outcome/aggregation) [21,34–36].

**Independent variables.** Based on the existing literature and the availability of the data in DHS, the factors taken into account in the analysis were classified into four categories: parental, health facility, obstetric, and child-related characteristics [17,21,35,37,38]. The analysis examined 11 parental factors such as the age of the mother (35–49, 20–34, 15–19 years),

residence (urban or rural), maternal and husband education (tertiary, none/primary, secondary), region (Nyanza, Eastern, Rift Valley, Nairobi, Coast, Central, Western Northeastern), maternal and husband working status (yes or no), and religion (Christian, Muslim, or others). The wealth index was generated from the 2022 KDHS data on household ownership of assets using the principal component analysis and grouped into 5 categories (poorest to richest). Whereas maternal autonomy was measured using two proxy variables: who heads the family (male or female) and who makes healthcare decisions for the mother/participant (self, partner, together with partner or someone else, or others). We included ten health facility, obstetric, and child-related factors in the analysis. Obstetric factors comprised pregnancy wantedness (no or yes), ages at first birth in 5-year age groups, antenatal (ANC) visits ($\geq 4$ or $\leq 3$), mode of delivery (cesarean or vaginal), the birth interval in months ($\leq 24$ or $\geq 25$) and pregnancy losses ($\geq 1$ or 0). Child factors included the sex of the child (male or female). Lastly, health facility factors included the specific place of childbirth (private, public, clinics, faith-based organizations, and Non-governmental Organizational health facilities). The analysis considered minutes to a healthcare facility of birth ($\leq 30$, 31–60, $\geq 61$) as a proxy indicator for access to the facility.

## Statistical analysis

Before analyzing data were cleaned. Descriptive statistics such as percentages were generated at the univariate level for all the categorical variables. We computed univariate logistic regression analyses, to find independent factors related to neonatal care quality. To investigate the factors correlated with newborn care quality while adjusting for other variables, any variables with P-values below 0.05 were added in a simple multivariate logistic regression model. The confidence intervals corresponding to 95% for all odds ratios are presented. The data was analyzed using SPSS's complex samples module (V29), which accounted for the complex sample design present in DHS data.

The complex sample package provides accurate parameter estimates by accounting for clustering, weighting, and sample stratification that occurred during the study participants' selection [39,40]. The sample weights were also added to all frequencies calculated to ensure the representativeness of the study results and account for unequal probability sampling across various strata [39]. A variance inflation factor (VIF) of less than 10 was utilized as a criterion to test multi-collinearity among all predictor variables in the model, during preliminary analyses [39,40].

## Ethical consideration

The Institutional Review Board of the ICF gave ethical clearance to conduct the 2022 KDHS. The Kenya National Bureau of Statistics conducted the study in partnership with other stakeholders. Given that the secondary data was publicly available, no ethical authorization was required to investigate it. However, MEASURE DHS provided access to the 2022 KDHS datasets. (https://www.dhsprogram.com/data/available-datasets.cfm). Adult participants provided written informed permission, as did legally designated representatives of minor participants.

## Results

### Demographic characteristics of the mothers of the newborn babies

DHS data about the quality of newborn care was obtained from 11,863 women who had recently given birth within five years preceding the survey (Table 1). The majority (74.4%) of women were aged 20 to 34, were from Eastern, Nairobi, Rift Valley, Central, and Nyanza provinces (76.7%), and lived in rural regions (61.4%). Moreover, 40% belonged to the richer

**Table 1. Demographic characteristics of the mothers of the newborn babies (Respondents).**

| Variable (N = 11,863) | n (weighted %) |
|---|---|
| **Maternal Age (years)** | |
| 15–19 | 785 (6.6) |
| 20–34 | 8825 (74.4) |
| 35–49 | 2253 (19) |
| **Region/Province** | |
| Nairobi | 1371 (11.6) |
| Northeastern | 406 (3.4) |
| Coast | 1107 (9.3) |
| Western | 1253 (10.6) |
| Eastern | 1336 (11.3) |
| Central | 1380 (11.6) |
| Nyanza | 1406 (11.8) |
| Rift Valley | 3605 (30.4) |
| **Residence** | |
| Rural | 7289 (61.4) |
| Urban | 4574 (38.6) |
| **Mother's working status/occupation** | |
| Not working | 5063 (42.7) |
| Working | 6791 (57.3) |
| **Maternal Education** | |
| Tertiary | 2321 (19.6) |
| Secondary | 4231 (35.7) |
| None/primary | 5311 (44.8) |
| **Wealth index** | |
| Richest | 2695 (22.7) |
| Richer | 2510 (21.2) |
| Middle | 2074 (17.5) |
| Poorer | 2062 (17.4) |
| Poorest | 2523 (21.3) |
| **Religion** | |
| Muslims | 1120 (9.7) |
| Christians | 10220 (88.5) |
| Others | 209 (1.8) |
| **Sex of household head** | |
| Female | 3380 (28.5) |
| Male | 8483 (71.5) |
| **Husband education** | |
| Tertiary | 2329 (24.5) |
| Secondary | 2984 (31.4) |
| None/primary | 4206 (44.2) |
| **Husband's working status** | |
| Not working | 858 (9) |
| Working | 8632 (91) |
| **Health seeking decision making** | |
| Joint | 4279 (44.9) |
| Partner | 1590 (16.7) |
| Self | 3618 (38.5) |

(*Continued*)

**Table 1.** (Continued)

| Variable (N = 11,863) | n (weighted %) |
|---|---|
| Others | 32 (0.3) |
| **Age at first birth (years)** | |
| ≥35 | 57 (0.5) |
| 30–34 | 229 (2.0) |
| 25–29 | 1215 (10.4) |
| 20–24 | 4630 (39.8) |
| ≤19 | 5503 (47.3) |
| **ANC Visits** | |
| ≤3 | 3157 (32.8) |
| ≥4 | 6472 (67.2) |
| **Pregnancy wanted** | |
| No | 1041 (8.8) |
| Yes | 10823 (91.2) |
| **Preceding pregnancy interval (months)** | |
| ≥25 | 6475 (75.4) |
| ≤24 | 2111 (24.6) |
| **Place of delivery** | |
| FBO | 574 (5.4) |
| NGO | 31 (0.3) |
| Private health facility | 1796 (16.9) |
| Public health facility | 6927 (65.2) |
| Home | 1246 (11.7) |
| Others | 55 (0.5) |
| **Mode of last delivery** | |
| Vaginal | 9043 (82.9) |
| Cesarean section | 1867 (17.1) |
| **Time to the health facility (minutes)** | |
| ≥61 | 496 (8) |
| 31–60 | 1168 (18.9) |
| ≤30 | 4515 (73.1) |
| **Sex of child** | |
| Female | 5121 (49.0) |
| Male | 5333 (51.0) |
| **Pregnancy losses** | |
| ≥1 | 2723 (23.0) |
| None | 9140 (77.0) |

**Note.** This table includes the demographic characteristics of all women who responded to the newborn care survey (N = 11,863).

(21.2%) and richest (22.7%) quintiles. The majority were Christian (88.5%), had completed at most secondary education (80.5%), and were working (57.3%). Furthermore, the majority of the participants' spouses were employed (91%), and 55.9% had completed their secondary education. The majority of participants (71.5%) lived in male-headed households and decided to seek health care services after consulting their spouses or someone else (44.9%). Most participants first gave birth while they were under 24 years old (87.1%), had a pregnancy interval of not less than two years following their previous delivery (75.4%), desired to have the most

**Table 2. Components of newborn care package received.**

| Variable | N | Weighed % (95% CI) |
|---|---|---|
| **Quality of newborn care (overall)** | 1503 | 32.7 (31.0–34.5) |
| Baby placed on mother's breast within 1 hour after birth | 3267 | 59.8 (57.7–61.8) |
| Delivered by a skilled health provider | 9463 | 90.4 (89.6–91.2) |
| Examined the cord | 3811 | 78.1 (76.6–79.5) |
| Showed how to clean the cord | 3609 | 73.9 (72.3–75.5) |
| Measured temperature | 3515 | 72.0 (70.3–73.6) |
| Advise on a baby who needs medical attention | 3205 | 65.6 (63.8–67.4) |
| Breastfeeding counselling | 3788 | 77.6 (76.1–79.0) |
| Observed breastfeeding | 3704 | 75.8 (74.3–77.3) |
| Weighing the baby | 4717 | 88.2 (87.1–89.3) |

**Note.** The table is based on participants who responded to specific questions related to newborn care (individual care service received). The participant responses varied depending on the newborn care service received.

recent pregnancy (91.2%), recorded at most four total antenatal visits (67.2%), and lived within 30 minutes from the health facilities (73.1%). Most mothers gave birth to newborn babies by vaginal birth (82.9%), and at public health facilities (65.2%). About half of neonates were of a male sex (51%). Additionally, about 23% of the participants reported at least a pregnancy loss (Table 1).

## Quality of newborn care

Overall, 32.7% of the mothers reported that their newborns had received all components of neonatal care (quality newborn care) after childbirth (Table 2). Among the components of newborn care, 90.4% of the mothers reported that their newborns were delivered by a skilled health provider, 78.1% had their newborn's umbilical cord examined by a health care worker, 88.2% were weighed, and had their temperature taken (72%). In addition, 59.8% of neonates were placed on the mother's breast by a medical professional within the first hour of birth. Participating mothers were instructed on ways to clean their newborn's umbilical cords (73.9%), observed during breastfeeding by medical professionals (75.8%), and given breastfeeding counseling (77.6%) or information about danger signs related to newborn sickness (77.6%).

## Factors associated with the quality of newborn care

Table 3 summarizes the factors associated with the quality of neonatal care in logistic regression analysis. In multivariate logistic regression, mothers who spent an average of one hour accessing the health facilities compared with those who spent less than half an hour were 1.33 (95%CI: 1.01–1.75) times more likely to report that their newborns had received quality newborn care. Mothers who gave birth in a non-government organization health facility were 30.37 (95%CI: 2.69–343.20) times more likely to report that their newborns had received quality newborn care compared with those who delivered from a faith-based organization.

On the contrary, in terms of regions, mothers who lived in Nyanza, Eastern, and Rift Valley provinces compared with those who lived in the coastal regions were 0.53 (95%CI: 0.34–0.82), 0.61 (95%CI: 0.39–0.94), and 0.62(95%CI: 0.41–0.93) times less likely to report that their newborns had received quality newborn care, respectively. Mothers who subscribed to other religions or faith (0.28 (95%CI: 0.10–0.76) compared with those from the Christian faith, were less likely to report that their newborns had received quality newborn care. Finally, mothers who gave birth via cesarean section were 0.44 (95%CI: 0.32–0.61) times less likely to report that their babies received quality newborn care than those who gave birth via spontaneous vaginal delivery.

**Table 3. Factors associated with the quality of neonatal care received by neonates of mothers aged 15–49 years.**

| Variable | Quality Neonatal care (N = 4601) | | CORs (95%CI) | P-value | AORs(95%CI) | p-value |
|---|---|---|---|---|---|---|
| | Yes *n (%)* | No *n (%)* | | | | |
| **Maternal Age (years)** | | | | 0.482 | | |
| 15–19 (Ref.) | 93 (2.0) | 229 (5.0) | 1 | | - | |
| 20–34 | 1155 (25.1) | 2358 (51.3) | 1.20 (0.87–1.66) | | - | |
| 35–49 | 255 (5.5) | 510 (11.1) | 1.23 (0.86–1.74) | | - | |
| **Residence** | | | | 0.475 | - | |
| Rural (Ref.) | 924 (20.1) | 1967 (42.8) | 1 | | - | |
| Urban | 580 (12.6) | 1129 (24.5) | 1.09 (0.92–1.31) | | - | |
| **Region/Province** | | | | 0.001 | | 0.003 |
| Coast (Ref.) | 156 (3.4) | 274 (6.0) | 1 | | - | |
| Northeastern | 25 (0.5) | 115 (2.5) | 0.92 (0.53–1.60) | | 0.57 (0.32–1.01) | |
| Eastern | 169 (3.7) | 385 (8.4) | 0.68 (0.44–1.04) | | **0.61 (0.39–0.94)** * | |
| Central | 185 (4.1) | 327 (7.1) | 0.89 (0.55–1.44) | | 0.99(0.60–1.63) | |
| Rift Valley | 410 (8.9) | 978 (21.2) | 0.82 (0.54–1.24) | | **0.62(0.41–0.93)** * | |
| Western | 198 (4.3) | 277 (6.0) | 1.24 (0.78–1.96) | | 0.89(0.57–1.43) | |
| Nyanza | 166 (3.6) | 379 (8.2) | **0.56 (0.35–.87)** * | | **0.53 (0.34–0.82)** * | |
| Nairobi | 193 (4.2) | 362 (7.9) | 0.69 (0.36–1.33) | | 0.59 (0.32–1.11) | |
| **Maternal Education** | | | | 0.003 | | 0.466 |
| None/primary (Ref.) | 590 (12.8) | 1445 (31.4) | 1 | | 1 | |
| Secondary | 595 (12.9) | 1104 (24.0) | **1.32 (1.10–1.59)** | | 1.15(0.88–1.49) | |
| Tertiary | 319 (6.9) | 547 (12.0) | **1.43 (1.10–1.85)** | | 1.26 (0.81–1.96) | |
| **Wealth index** | | | | <0.001 | | 0.332 |
| Poorest (Ref.) | 216 (4.7) | 734 (15.9) | 1 | | 1 | |
| Poorer | 275 (5.9) | 551 (12.0) | **1.69 (1.35–2.13)** | | 1.18 (0.85–1.63) | |
| Middle | 302 (6.6) | 517 (11.2) | **1.99 (1.59–2.49)** | | 1.16 (0.83–1.62) | |
| Richer | 398 (8.7) | 625 (13.6) | **2.17 (1.70–2.77)** | | 1.31(0.89–1.91) | |
| Richest | 312 (6.8) | 671 (14.6) | **1.59 (1.21–2.08)** | | 0.91 (0.58–1.41) | |
| **Religion** | | | | 0.009 | | 0.040 |
| Christians | 1349 (30.1) | 2650 (59.1) | 1 | | 1 | |
| Muslims | 117 (2.6) | 303 (6.7) | 0.76 (0.55–1.05) | | 1.08 (0.65–1.79) | |
| Others (Ref.) | 10 (0.2) | 58 (1.3) | **0.34 (0.17–0.68)** * | | **0.28 (0.10–0.76)** * | |
| **Mother's Working status/occupation** | | | | 0.025 | | 0.665 |
| Not working (Ref.) | 591 (12.9) | 1363 (29.7) | 1 | | 1 | |
| Working | 910 (19.8) | 1731 (37.6) | **1.21 (1.03–1.44)** * | | 1.05 (0.84–1.31) | |
| **Husband Education** | | | | 0.004 | | 0.490 |
| None/primary (Ref.) | 466 (12.8) | 1130 (31.0) | 1 | | 1 | |
| Secondary | 393 (10.7) | 766 (21.0) | **1.24 (1.02–1.51)** * | | 1.07 (0.82–1.40) | |
| Tertiary | 342 (9.3) | 557 (15.2) | **1.490 (1.162–1.910)** * | | 1.28 (0.85–1.92) | |
| **Husband's Working status/occupation** | | | | <0.001 | | 0.240 |
| Not working (Ref.) | 59 (1.6) | 247 (6.8) | 1 | | 1 | |
| Working | 1134 (31.1) | 2203 (60.5) | **2.15 (1.64–2.81)** | | 1.23 (0.87–1.74) | |
| **Sex of Household head** | | | | 0.668 | | |
| Male (Ref.) | 1074 (23.3) | 2187 (47.5) | 1 | | - | |
| Female | 430 (9.4) | 910 (19.8) | 0.96 (0.81–1.15) | | - | |
| **Health seeking decision making** | | | | 0.014 | | 0.557 |
| Self (Ref.) | 490 (13.4) | 926 (25.4) | 1 | | 1 | |

*(Continued)*

**Table 3.** (Continued)

| Variable | Quality Neonatal care (N = 4601) | | CORs (95%CI) | P-value | AORs(95%CI) | p-value |
|---|---|---|---|---|---|---|
| | Yes n (%) | No n (%) | | | | |
| Partner | 143 (3.9) | 421 (11.5) | **0.64 (0.49–0.84)** * | | 0.81 (0.58–1.14) | |
| Joint | 563 (15.4) | 1099 (30.2) | 0.97(0.79–1.19) | | 1.02 (0.79–1.31) | |
| Others | 0 (0) | 7 (0.2) | 1.24 0.29–5.42 | | 1.15 (0.19–7.12) | |
| **ANC Visits** | | | | 0.002 | | 0.443 |
| ≤3 (Ref.) | 418 (9.1) | 1046 (22.7) | 1 | | 1 | |
| ≥4 | 1085 (23.6) | 2051 (44.6) | **1.32 (1.11–1.58)** * | | 1.09 (0.86–1.39) | |
| **Age at first birth (years)** | | | | 0.589 | | |
| ≤19 (Ref.) | 707 (15.3) | 1499 (32.6) | 1 | | - | |
| 20–24 | 602 (13.1) | 1236 (26.9) | 1.03 (0.86–1.24) | | - | |
| 25–29 | 157 (3.4) | 307 (6.7) | 1.09 (0.79–1.49) | | - | |
| 30–34 | 33 (0.7) | 52 (1.1) | 1.34 (0.74–2.44) | | - | |
| ≥35 | 5(0.1) | 3 (0.1) | 3.10 (0.81–11.81) | | - | |
| **Preceding pregnancy interval (months)** | | | | 0.051 | | 0.154 |
| ≤24 (Ref.) | 223 (6.7) | 562 (16.8) | 1 | | 1 | |
| ≥25 | 860 (25.7) | 1701 (50.8) | **1.273(0.99–1.62)** | | 1.20 (0.93–1.55) | |
| **Pregnancy wanted** | | | | 0.239 | | |
| No (Ref.) | 144 (3.1) | 255 (5.6) | 1 | | - | |
| Yes | 1360 (29.6) | 2841 (61.7) | 0.85 (0.65–1.11) | | - | |
| **Mode of last delivery** | | | | <0.001 | | <0.001 |
| Vaginal (Ref.) | 1315 (28.6) | 2494 (54.2) | 1 | | 1 | |
| Cesarean section | 188 (4.1) | 603 (13.1) | **0.59 (0.45–0.78)** | | **0.44 (0.32–0.61)** * | |
| **Place of delivery** | | | | <0.001 | | <0.001 |
| FBO (Ref.) | 103 (2.5) | 167 (4.0) | 1 | | 1 | |
| Public health facility | 1118 (26.9) | 1906 (45.9) | 0.95 (0.63–1.42) | | 1.06 (0.64–1.79) | |
| Private health facility | 257 (6.2) | 561 (13.5) | 0.74 (0.46–1.19) | | 0.95 (0.50–1.79) | |
| NGO | 11 (0.3) | 5 (0.1) | 3.40 (0.39–29.51) | | **30.37 (2.69–343.20)** * | |
| Others | 3 (0.1) | 20 (0.5) | 0.22 (0.041–1.13) | | 0.30 (0.06–1.60) | |
| **Pregnancy losses** | | | | 0.569 | | |
| None (Ref.) | 1319 (28.7) | 2694 (58.5) | 1 | | - | |
| ≥1 | 184 (4.0) | 403 (8.8) | 0.94- (0.74–1.18) | | - | |
| **Time to the health facility (minutes)** | | | | 0.006 | | 0.125 |
| ≤30 (Ref.) | 293 (6.4) | 584 (12.7) | 1 | | 1 | |
| 31–60 | 1124 (24.4) | 2236 (48.6) | 1.00 (0.83–1.21) | | **1.33 (1.01–1.75)** * | |
| ≥61 | 86 (1.9) | 277 (6.0) | **0.62 (0.46–0.83)** * | | 1.09 (0.74–1.60) | |
| **Sex of child** | | | | 0.736 | | |
| Female (Ref.) | 734 (16.0) | 1535 (33.4) | 1 | | - | |
| Male | 769 (16.7) | 1562 (33.9) | 1.03 (0.89–1.22) | | - | |

*Note.*—not evaluated in that model

* = significant at 0.05, **Bold** = significant, CI = confidence interval, NGO = nongovernmental organization, ANC = antenatal care, FBO = faith-based organizations, AORs = adjusted odds ratios, CORs = crude odds ratios, Ref. = reference category. This table is based on all participants who responded to all nine questions related to newborn care (N = 4601). Participants with missing figures or those who responded to some questions were eliminated from the analysis.

## Discussion

The study determined the factors associated with the quality of newborn care in Kenya using the 2022 KDHS. Overall, 32.7% of the mothers reported that their newborns had received all components of neonatal care (quality newborn care) after childbirth. These results are lower compared with the overall quality of neonatal care reported in Ethiopia at 66.3% and 38.4% [41,42]. Among the components of neonatal care, 90.4% of the newborns were delivered by a skilled health provider. This is higher than 86% recorded globally [43], 79.2%, and 51.8% recorded in Benin and Ethiopia, respectively [44,45]. However, 78.1% had their newborn's umbilical cord examined by a healthcare worker which is similar to 75.4% and 70.3% recorded in Ethiopia and Senegal respectively [41,46]. Seventy three percent (73.9%) of participating mothers were shown how to clean their umbilical cord higher than 66% recorded in Uganda [47]. Additionally, 59.8% of the newborns were placed on the mother's breast within 1 hour after birth by a healthcare worker, higher than the global rate of 45% [48], 29.4%, and 47.8% recorded in Senegal and India respectively [46,49].

In this study, 75.8% of mothers were observed while breastfeeding their newborns by health care professionals, while 77.6% received breastfeeding counseling which is higher compared to 31.9% recorded in a study by Hassounah et al. [50]. In this study, 77.6% of mothers were counseled about the danger signs associated with newborn illness (77.6%), 88.2% of their newborns were weighed and 72% had their temperature taken. In sum, although the overall quality of newborn care reported by the mothers appears low, statistics about specific components of that care paint a seemingly positive picture of a better quality of newborn care in Kenya when compared with previous studies, which can be attributed to several initiatives by the government of Kenya to improve maternal and child health care such as the free maternal and newborn care policy [51,52]. This also means that various stakeholders may need to set policy targets related to newborn care based on all components of newborn care and not a few.

Mothers who lived in Nyanza, Eastern, and Rift Valley provinces compared with those who lived in the coastal regions were less likely to report that their newborns had received quality newborn care. This study's findings about the three counties could be supported by the fact that these are underdeveloped counties that were considered disadvantaged regions by Kenya's presidential initiatives for improvement of the quality of maternal and child health services [51–53]. Such areas may not have attained quality health care services thus explaining the poor quality of neonatal care received. Such findings are also consistent with a study in Ethiopia that found the marginalization (disregard) of neonatal health services in rural and underdeveloped regions reduced the quality of neonatal health services provided in such areas [54]. Such regions need to be given priority when it comes to the distribution of newborn-related health care services by ministries of health in Kenya and other sub-Saharan countries.

Consistent with literature, mothers who subscribed to other religions or faith compared with those from the Christian faith, were less likely to report that their newborns had received quality newborn care. One study observed that religion is a key predictor of neonatal health outcomes [55]. This could be explained by the positive impact of some religions as documented by Brelsford et al. [56] where some mothers were more likely to comply with newborn quality care practices as compared with other religions. This finding presents such denominations as a platform for presenting newborn health-related information to the populations through sensitization.

Furthermore, mothers who gave birth through cesarean section were less likely to report that their newborns had received quality newborn care when compared with those who gave birth through spontaneous vaginal delivery. This could be because mothers after cesarean section are less physically functioning as compared with those who have a safe vaginal delivery

and hence are unable to cooperate with health workers fully in initiating interventions like timely breastfeeding and thermoregulation of neonates compromising the quality of neonatal care [57]. Ceasarian section has previously been associated with higher incidences of neonatal complications and mortality due to pregnancy-related conditions that may or may not have necessitated the ceasarian section [58]. Therefore, the level of care needed by such babies may not be easily realizable in low-resource settings of Kenya, as compared with those that deliver vaginally. This calls for capacity building of health professionals by government and nongovernment stakeholders in delivering advanced neonatal interventions in facilities authorized to perform cesarian sections.

Mothers who gave birth in non-government organization health facilities were more likely to report that their newborns had received quality newborn care compared with those who delivered from faith-based organizations. This finding concurs with other studies that found that public and private health facilities lacked essential supplies and equipment needed to provide quality newborn care. Those facilities also lacked basic and effective aspects of essential newborn care like skin-to-skin contact between mother and babies as compared with other facilities [59,60]. Scholars have highlighted the important role played by NGOs in providing quality health care and the need for government authorities to seek collaborations with such organizations [61,62]. NGO faciltities emphasize continuity of quality care and provide affordable services as compared with other facilities, thus this necessitates governement authorities to facilitate a good working environment for their operations [63]. NGOs also provide quality healthcare since they focus their resources delivering on healthcare outcomes unlike governments that may focus on divergent political and social agenda when providing healthcare [64].

Mothers who spent an average of one hour accessing the health facilities compared with those who spent less than half an hour were more likely to report that their newborns had received quality newborn care. Unlike in this study, increased travel time to the delivery facility has been associated with an increased risk of neonatal mortality and complications [65]. Therefore, we recommend further qualitative studies by researchers to explore why mothers nearest the health facilities were more likely to report low-quality newborn care compared with those farther away from the health facilities. In sum, this study seems to suggest that facility-related and parental social factors are associated with receiving quality newborn care.

### Study strengths and limitations

Data from the Kenya Demographic Health Survey 2022 was weighted and is nationally representative, therefore our results can be generalizable to all newborns in Kenya. Furthermore, DHS is known for high response rates and consistent quality data collection techniques across states, thus our results are comparable to studies conducted elsewhere of the same design. However, because this study used a cross-sectional design, its conclusions are imprecise in demonstrating causality, and there is a possibility of recall bias from the participants because this study was based on self-reported data. Finally, due to the large number of missing data points, we included fewer participants in univariate and multivariate logistic regression analyses. Despite its limitations, this study presents useful information on factors influencing the quality of newborn care in Kenya.

### Conclusions

This study found the overall quality of newborn care to be low (below average), with some vital components of care scoring below the UN projected targets for SDG 3. These results highlight a gap in neonatal health care provision hence the need for responsible stakeholders to take action. This study also seems to suggest that facility-related and parental social factors are

associated with receiving quality newborn care. The study findings also show very low quality of neonatal care in some counties of Kenya which prompts governments in subsaharan Africa to prioritize universal health coverage hence even distribution of neonatal health services to such marginalized areas to improve accessibility and quality of neonatal health services. Emphasis should also be put on newborns delivered by the ceasarian section since they are at high risk of receiving low-quality newborn care. Therefore, this calls for capacity building of health professionals by government and nongovernment stakeholders in delivering advanced neonatal interventions in facilities authorized to perform cesarian sections. Government authorities and other stakeholders should also target religious denominations as a platform for sensitizing citizens about neonatal health care services. Collaborations between government authorities and NGO health facilities or newborn centers of excellence could also facilitate the provision of quality newborn care.

## Acknowledgments

We thank the DHS program for making the data available for this investigation.

## Author Contributions

**Conceptualization:** John Baptist Asiimwe, Lilian Nuwabaine.

**Data curation:** John Baptist Asiimwe.

**Formal analysis:** John Baptist Asiimwe, Earnest Amwiine, Angella Namulema, Quraish Sserwanja, Joseph Kawuki, Mathius Amperiize, Imelda Namatovu, Lilian Nuwabaine.

**Investigation:** John Baptist Asiimwe, Earnest Amwiine, Shamim Nabidda.

**Methodology:** John Baptist Asiimwe, Earnest Amwiine, Angella Namulema, Quraish Sserwanja, Joseph Kawuki, Mathius Amperiize, Shamim Nabidda, Imelda Namatovu, Lilian Nuwabaine.

**Software:** John Baptist Asiimwe, Quraish Sserwanja, Shamim Nabidda, Lilian Nuwabaine.

**Validation:** John Baptist Asiimwe, Joseph Kawuki, Imelda Namatovu, Lilian Nuwabaine.

**Visualization:** John Baptist Asiimwe, Joseph Kawuki, Mathius Amperiize, Shamim Nabidda, Lilian Nuwabaine.

**Writing – original draft:** John Baptist Asiimwe, Earnest Amwiine, Angella Namulema, Quraish Sserwanja, Joseph Kawuki, Mathius Amperiize, Shamim Nabidda, Imelda Namatovu, Lilian Nuwabaine.

**Writing – review & editing:** John Baptist Asiimwe, Earnest Amwiine, Angella Namulema, Quraish Sserwanja, Joseph Kawuki, Mathius Amperiize, Shamim Nabidda, Imelda Namatovu, Lilian Nuwabaine.

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
