## [Decision Letter · Decision Letter 0]

10 Sep 2024

PGPH-D-24-01885

Quality of newborn care and associated factors: An analysis of the 2022 Kenya demographic and health survey.

Dear Dr. Asiimwe

Thank you for submitting your manuscript to PLOS Global Public Health. After careful consideration, we feel that it has merit but does not fully meet PLOS Global Public Health’s publication criteria as it currently stands. Therefore, we invite you to submit a revised version of the manuscript that addresses the points raised during the review process. Please submit your revised manuscript by 9 November 2024. If you will need more time than this to complete your revisions, please reply to this message or contact the journal office at globalpubhealth@plos.org. Please include the following items when submitting your revised manuscript:

We look forward to receiving your revised manuscript.

Kind regards,

Mohammad Shahidul Islam, PhD

Academic Editor

Journal Requirements:

Additional Editor Comments (if provided):

Reviewer 1:

“Quality of newborn care and associated factors: An analysis of the 2022 Kenya demographic and health survey.” by Asiimwe JB et al., a manuscript on a very important aspect regarding newborn care – to identify the factors associated with the quality of newborn care in Kenya.

The world has made substantial progress in child survival since 1990. However, the decline in neonatal mortality has been slower than that of post-neonatal under-5 mortality. Plans to improve newborn survival should be built on a strong foundation of essential newborn care and align with Every Newborn Action Plan (ENAP) and Ending Preventable Maternal Mortality (EPMM) targets on antenatal care, postnatal care, skilled health personnel and emergency obstetric and newborn care.

The authors have tried to find out the condition of the factors that influence the quality of newborn care in Kenya using data from nationally representative survey¸ Kenya Demographic and Health Survey (KDHS) 2022. They have several recommendations from the findings for better ensuring the accessibility and quality of neonatal health services. I believe identifying the gaps in quality of newborn care through this publication and taking appropriate measures will help to improve neonatal care in in Kenya and elsewhere in developing countries.

Although it is well written, I have few observations that need to be addressed before publication, as mentioned below:

Line 66: ‘a third of the neonates’ is fine, ‘(1 in 3)’ probably is not necessary.

Line 100: Here emphasis has been given on ‘delayed initiation of breast feeding’. Early initiation of breastfeeding, within one hour of birth, protects the newborn from acquiring infection and reduces newborn mortality. It seems to me that his sentence does not well fit here, also does not go with the next sentence. It could be an important finding if data was there about the time of initiation of breast feeding.

Line 119: Here mentioned ‘some studies’, but reference is given for one study [31].

Line 210: ‘Most participants gave birth while they were under 24 years old (87.1%)'. It is the information about the age at first birth (Table 1), not of the delivery that was surveyed, this should be clarified.

Line 216: About 1 in 10 women will have a miscarriage over a lifetime, according to a series of articles in The Lancet. Is 23% fewer pregnancy loss? Does this figure match with data given in Table 3? Probably there is many missing data, need comments on that.

Table 3: I have confusion about the numbers of cases, I cannot match them. What is the denominator?

Line 303-35: Babies born by ceasarian section might have higher incidences of neonatal complications and mortality probably due the indication that necessitated the ceasarian section. There is no reason that such babies are prone to adverse conditions like ‘concomitant diarrhea and acute respiratory infections’. The first publication in the reference mentioned babies aged less than 6 months and the second publication is on a special situation.

Line 355: ‘Abbreviations’ are probably not required separately. The rule is to spell out the full term in text at its first mention, its abbreviation in parenthesis and the abbreviation from then on.

Gestational age (term or preterm) and birth weight (AGA or LBW) is a is major determinant of neonatal morbidity and mortality. I am not sure whether this information was available for analysis.

Reviewer 2

Many thanks for your work on this critical subject matter - Quality of newborn care and associated factors.

Generally the method using a 2-stage stratified sampling across 8 regions in Kenya is a valid and interesting topic. However, upon delving into this manuscript I found a number of improvements that would make it more convincing and stronger.

Generally, the construction of "Quality" care has been fairly difficult to pin down - I wonder why the WHO framework (2016) for quality of newborn care was not relied upon to define the provision of quality care. Instead of or in addition to REF3 for recommendations for "a positive childbirth experience." OR another more sensible proximal to distal modality (bodily, maternal, delivery, after care, parental, SES/demographic) to list the risk factors impinging on newborn quality care provision. Regardless, the risks are presented here and compared with other previous work in the field so it just begs for a more hard definition. Also, please remember that many people are not aware with which regions of Kenya are marginalized or not (ln 69).

There was another statistical oddity that I thought undermined the paper's credibility and should be revised. In table 1 Religions are listed as Christian, Muslim and Others. I see that the regression was performed using "Other" as the nominal or base and then concluding that both Christian and Muslim religions seem to be protective and encourage quality of care. Where it would be more convincing to me that the "other" religions were in fact the marginalised groups (ln69). This is exacerbated by the number of Others (209 1.8%) and it is generally not good practice to use a small and under-representative group as the base to a regression is not suitable.

I was quite thrown by the caesarian findings as well, these must take place in a facility, and it was odd to see that many shortfalls related to quality happen there, including failing to weigh the baby?

You mention that dummy variables were constructed but do not specify which or how these composite variables are created or compared. it is assumed that functionally this would happen in line, so either define these or omit the dummy vars (178)

Your introduction seems to list a number of risks and such from various places, but does not clearly assert that you will be defining your risks as X based on Y or Y+Z. Thus the reader is left guessing until the line 147 Dependent variables section or Table 1 essentially. Consider a complete revision and tightening up.

In fact, the conclusions seem quite foregone, that quality risk comes from facilities and parental factors.

Here are some additional comments to crosscheck and address.

1. Stata version seems to be out of date (ln 44)

2. ln 55 - add "based on the above definitions" to the end of the sentence to clarify

3. 183 typo Odd"s" ratio.

4. ln 79 - Very awkward wording revise

5. ln 80 - change in to "to"

6. ln 81 - change "thus, reducing" to "and can reduce"

7. ln 117 - consider change "monitor the quality" to "invest in monitoring the quality"

8. ln 148 - consider revision to clarify method of aggregate variable creation here. is it a binary in case of total quality 1/0 or some other cut-off - may also consider nodding to other risk scoring efforts somewhere.

9. ln 160 - you list 4 categories but say 3.

10. ln 213 - 30mins - needs space

11. ln 228 remove "babies"

12. ln 260 - respective to what, need a pre-list

13. ln 271 - the title of ref 48 does not specify where

14. ln 291 - reference is from Nepal, perhaps not apt.

15. ln 340 - informal voice – remove

Editor Comments:

The submitted manuscript titled "Quality of Newborn Care and Associated Factors: An Analysis of the 2022 Kenya Demographic and Health Survey" explores the quality of newborn care in Kenya. The findings of this study is particularly valuable for many LMICs, where essential newborn care is often difficult to access. The paper identifies several barriers to improving newborn care in Kenya, which may also be relevant in other regions with high neonatal mortality rates. However, the reviewers have noted some weaknesses/limitations in the manuscript that must be addressed adequately before it can be considered for publication.

---

## [Editor Report · Decision Letter 1]

9 Oct 2024

PGPH-D-24-01885R1

Quality of newborn care and associated factors: An analysis of the 2022 Kenya demographic and health survey.

Dear Dr.Asiimwe,

Thank you for submitting your manuscript to PLOS Global Public Health. After careful consideration, we feel that it has merit but does not fully meet PLOS Global Public Health’s publication criteria as it currently stands. Therefore, we invite you to submit a revised version of the manuscript that addresses the points raised during the review process.

The revised version of the manuscript addresses most of the concern raised by the reviewers. However, we suggest to reconsider the concern raised by the second reviewer regarding the use of 'other religion' as the reference category in Table 3 to estimate the impact of religion on receiving quality newborn care. Due to the smaller sample size in the other religion category, this choice may undermine the credibility of the paper, and the interpretation should be revised accordingly. Additionally, as pointed out by Reviewer 1, in Table 3, please mention the total number of participants analyzed to ensure readers can understand how the percentages were calculated.

We look forward to receiving your revised manuscript.

Kind regards,

Mohammad Shahidul Islam, PhD

Academic Editor

Journal Requirements:

Additional Editor Comments (if provided):

The revised manuscript has addressed most of the reviewers' concerns. However, the authors should reconsider the concern raised by the second reviewer regarding the use of 'other religion' as the reference category in Table 3 to estimate the impact of religion on receiving quality newborn care. Due to the smaller number of participants in this category, this choice may undermine the credibility of the paper, and the interpretation should be revised accordingly.

Additionally, as pointed out by Reviewer 1, in Table 3, please mention the total number of participants analyzed to ensure readers can understand how the percentages were calculated.
---

## [Editor Report · Decision Letter 2]

15 Oct 2024

PGPH-D-24-01885R2

Quality of newborn care and associated factors: An analysis of the 2022 Kenya demographic and health survey.

Dear Dr. Asiimwe

Thank you for submitting your manuscript to PLOS Global Public Health. After careful consideration, we feel that it has merit but does not fully meet PLOS Global Public Health’s publication criteria as it currently stands. Therefore, we invite you to submit a revised version of the manuscript that addresses the points raised during the review process.

We look forward to receiving your revised manuscript.

Kind regards,

Mohammad Shahidul Islam, PhD

Academic Editor

Journal Requirements:

Additional Editor Comments (if provided):

Thank you for sharing the revised version of the manuscript. It adequately addresses the reviewers' concerns. However, in Table 3, the percentage calculations appear incorrect. It seems the authors calculated the percentage for each cell based on the total number of respondents. Additionally, some categories (e.g., mother's age) do not sum to 100%. Ideally, the percentages for each row (Yes and No) should add up to 100%. Please revisit the percentage calculations and correct if there is any errors.

---

## [Editor Report · Decision Letter 3]

17 Oct 2024

Quality of newborn care and associated factors: An analysis of the 2022 Kenya demographic and health survey.

PGPH-D-24-01885R3

Dear Asiimwe,

We are pleased to inform you that your manuscript 'Quality of newborn care and associated factors: An analysis of the 2022 Kenya demographic and health survey.' has been provisionally accepted for publication in PLOS Global Public Health.

Best regards,

Mohammad Shahidul Islam, PhD

Academic Editor
